# Affective Neuroscience Personality Scales and Early Maladaptive Schemas in Depressive Disorders

**DOI:** 10.3390/ijerph19138062

**Published:** 2022-06-30

**Authors:** Monika Talarowska, Grzegorz Wysiadecki, Jan Chodkiewicz

**Affiliations:** 1Institute of Psychology, Faculty of Educational Sciences, University of Lodz, Smugowa 10/12, 90-433 Łódź, Poland; grzegorz.wysiadecki@umed.lodz.pl (G.W.); jan.chodkiewicz@now.uni.lodz.pl (J.C.); 2Department of Normal and Clinical Anatomy, Medical University of Lodz, 90-419 Łódź, Poland

**Keywords:** ANPS, EMS, early maladaptive schemas, Affective Neuroscience Personality Scales, depressive disorders

## Abstract

Aim: The aim of this study was to assess the interrelationships of Young’s early maladaptive schemas with indicators of specific neural emotional systems conceptualized in Panksepp’s theory in a group of people suffering from depressive disorders. Materials and methods: The Affective Neuroscience Personality Scales (ANPS) v. 2.4. and J. Young’s Early Maladaptive Schema Questionnaire (YSQ-S3-PL) were used. Ninety (90) individuals aged 18–58, including 45 people treated for depression (DD group), were qualified to participate in the experiment. Results: The subjects in the DD group scored statistically significantly lower than the subjects from the control group (CG group) on the three ANPS scale domains, namely SEEKING, PLAY, and ANGER. The subjects with depressive symptoms scored significantly higher in the YSQ-S3-PL questionnaire on two domains of early maladaptive schemas, i.e., “Impaired autonomy and performance” and “Other-directedness”. Regression analysis results indicate that impairment of the emotional SEEKING system explains most of the variability in the following typical domains of depression: “Disconnection and rejection”, “Impaired autonomy and performance”, and “Other-directedness”. For score variability in the domain area of “Impaired limits”, the ANGER system was found to be most significant, and the FEAR system proved the same for “Overvigilance and Inhibition”. Conclusions: 1. Two domains of early maladaptive schemas are significant for the onset of depressive symptoms, namely “Impaired autonomy and performance” and “Other-directedness”, linked to difficulties in engaging in behaviors to meet one’s own needs. 2. Impairment of the neural emotional SEEKING system most significantly explains the variability in depression-typical areas of early maladaptive schemas.

## 1. Introduction

The theory of early maladaptive schemas (EMS), created by Jeffrey E. Young, seems to comprehensively explain the mechanism and, above all, the dynamics of mental disorders arising from early childhood experiences [1,2].

According to the assumptions of EMS, the experiences we make in our earliest stages of development shape relatively stable patterns of functioning and beliefs about ourselves, other people, and the surrounding world. These patterns are referred to as schemas. The primary source of dysfunctional schemas is the inability to meet or inadequately meet one (or more) of the child’s basic developmental needs (so-called core needs) [3,4]. Failure to meet these needs causes emotions that are difficult for the child, such as anxiety, anger, shame, or guilt; to avoid experiencing them, individuals engage in a variety of behavioral and coping strategies, which—while reducing tension—also contribute to the perpetuation of certain schemas [5]. Over the past few years, Young’s theory of schemas has become a focal point for many studies that have attempted to explain the etiology of numerous psychiatric disorders [6].

The achievements of modern medicine and neuroscience allow us to assume that biological factors play a significant role in the formation of human personality [7]. One theory that seeks to understand the neural foundations of emotion, and thus human personality, is the Affective Neuroscience Personality Theory (ANPT) by Panksepp [8]. According to Panksepp’s view, words (narratives) and environmental influences do not fully explain animal and human behavior. However, they may be explained by processes that originate in human brain activity [9].

Panksepp suggests the existence of six neural emotional, and motivational systems related to older subcortical structures of the mammalian brain that were developed by nature over millions of years of evolution as adaptations crucial to animals’ survival [10,11]. These systems are mostly homologous in all mammals. Furthermore, the neurochemistry of these systems is similar in all mammals [12]. According to the cited authors, in the face of external and internal environmental challenges, primary emotional systems allow for the rapid arousal and coordination of dynamic forms of brain organization. Emotions, in this view, are adaptive and innate, and the neural circuits associated with them have evolved to ensure that individuals are highly efficient at surviving and passing on genes to their offspring [13].

According to Panksepp, primary emotional processes, which are instinctive and represent a form of evolutionary “memory”, underpin the emotional and drive-related activity of the brain. This adaptation is essential for mammals to survive in the environment. Primary processes are associated with subcortical centers in the lower brain, mainly in its medial part [14]. Lower brain areas not only have an evolutionary advantage in creating primary emotions but also generate instinctive behavioral responses that are closely related to primary effects. According to Panksepp, emotions are an important foundation of personality, and personality assessment can serve as information regarding the aforementioned subcortical effects [15]. Additionally, differences in the response of primary emotional systems underlie human personality [16].

The location of specific neural systems was determined by observing the instinctive responses of animals that occur under electrical or chemical stimulation of particular brain regions [17]. The neural systems listed by Panksepp include SEEKING (interest), ANGER (rage), FEAR (anxiety), CARE (caring/nurturance), SADNESS/PANIC (separation distress/grief), and PLAY (playfulness/joy) [18,19]. Each of these systems can be activated by stimulating separate (although overlapping) areas of the brain. However, they usually work together to increase the adaptability of feelings, thoughts, and behaviors of the individual [20]. The location in the brain and characteristics of the neural emotional systems described by Panksepp are shown in Table 1.

Panksepp assumes that emotions are the basis of personality; hence the tendency to respond with a specific effect associated with the activation of the relevant neural emotional system can be associated with the formation of a specific response pattern or personality trait [9]. Moreover, the operation of these systems may help in understanding the etiology and course of many psychiatric disorders [21].

### Aim

Until the moment of compiling this article, the authors did not confirm the presence of other studies linking Jeffrey Young’s theory of early maladaptive schemas with J. Panksepp’s Affective Neuroscience Personality Theory. The study presented herein is, therefore, the first attempt to deal with this matter. It is hoped that the results obtained lead to a better understanding of the biological basis of early maladaptive schemas and, in the future, contribute to an increase in the effectiveness of applied therapeutic interventions.

The authors attempted to assess the relationship between early schemas and neural emotional systems in the depressive disorders group. These disorders were selected due to the frequency of their prevalence in the population and the magnitude of the clinical, social, and economic costs they generate [23].

Bearing in mind the aforementioned correlations, the aim of the presented study was to investigate and assess the interrelationships between J. Young’s early maladaptive schemas and indicators of individual emotional systems distinguished in J. Panksepp’s theory in a group of patients with depressive disorders.

## 2. Material

Ninety (90) individuals aged 18–58 took part in the study. The sample size was estimated using the G*Power program [24]. Forty-five (45) subjects in the study group had been diagnosed with a depressive disorder by a psychiatrist (depressive disorder group, DD group, F33) [25]. The individuals in the DD group were matched by taking into account the pharmacological treatment applied. Only the patients taking SSRI medications at the time of the study and who were receiving standard treatment with SSRIs were eligible to participate [26]. Subjects who had been taking SSRIs for no more than two weeks prior to the start of the study were eligible to participate. Only the subjects who scored above seven on the HDRS scale, indicating the presence of depressive symptoms, were qualified for the DD group. The mean severity of depressive symptoms as measured by the Hamilton Depression Rating Scale in the DD group was *M* = 25.07 (*SD* = 7.91), which indicates a significant intensification of symptoms of the disease.

The comparison group consisted of 45 healthy subjects (CG group) recruited by the snowball method, matched for gender and education. There were no statistically significant differences in gender between the groups analyzed (*chi^2^* = 0.498, *p* = 0.481), but there were statistically significant differences in age (*t* = 11.174, *p* < 0.001). The respondents also performed the BDI-II test to exclude people with symptoms of depression. In each case, the mental condition of the respondents was assessed by a psychiatrist. The social and demographic characteristics of the study group are presented in Table 2.

In the DD group, the subjects diagnosed with disorders other than axis disorders (F33) were excluded from the study.

### 2.1. Method

A self-reported survey, enabling the collection of sociodemographic data, as well as the following questionnaires, were used in the study:

J. Young’s early maladaptive schemas questionnaire (YSQ-S3-PL) in the Polish adaptation by [27].

The method examined the intensity of each of the 18 schemas based on a self-report of the respondent, who was asked to respond to the highlighted statements. This allows the pattern of schemas characteristic of the particular person to be identified. The questionnaire consists of 90 test items (five for each schema). The scores for each schema are in the range of 5 to 30. The arithmetic mean for each schema and the total score for all were also calculated. The Polish version of the method has acceptable psychometric properties [27].

When analyzing the results obtained in the YSQ-S3-PL questionnaire, the authors used the division into five schema areas (domains) distinguished by its author, namely: “Disconnection and rejection”, “Impaired autonomy and performance”, “Impaired limits”, “Other-directedness”, and “Overvigilance and inhibition”. These domains were distinguished by the degree of unmet needs that are crucial to normal human development (respectively), i.e., secure attachment to others; autonomy; realistic boundaries and self-control; freedom to express needs; and spontaneity and play [28].

#### 2.1.1. The Affective Neuroscience Personality Scales (ANPS) v. 2.4

The ANPS version 2.4 of 2004, developed by Davis et al. (2011), was used in the study. This tool is designed to assess endophenotypes associated with the activity in core emotional systems that emerged from research in affective neuroscience [18,19]. This scale, with the permission of its author, was translated into Polish by competent specialists.

The ANPS consists of 112 statements rated on a 4-point Likert scale (strongly disagree (1) to strongly agree (4)). The results represent the intensity of six emotional systems: SEEKING (interest), ANGER (rage), FEAR (anxiety), CARE (caring/nurturance), SADNESS/PANIC (separation distress/grief), and PLAY (playfulness/joy). Of those cited so far, LUST was excluded due to its lesser relevance in light of current conceptions of human personality [29,30,31]. Detailed characteristics of the neural emotional systems are shown in Table 1.

#### 2.1.2. Hamilton Depression Rating Scale (HDRS)

The 21-item Hamilton Depression Scale [32] was used to assess the severity of depression in the DD group. It consists of items that assess the degree of depressed mood; the presence of psychomotor retardation and/or inhibition; the severity of experienced guilt; the presence of sleep and/or appetite disturbances; the presence of anxiety symptoms; the presence of suicidal thoughts, tendencies, and attempts; and critical attitudes toward disease symptoms.

The intensity of depressive disorders was classified based on the grades distinguished in Demyttenaere et al. [33]: <7—no depressive symptoms; 8–12—mild intensity of depressive symptoms; 13–17—moderate intensity of depressive symptoms; 18–29—severe intensity of depressive symptoms; >30—highly severe intensity of depressive symptoms.

### 2.2. Study Procedure

The study presented herein was conducted between 2019 and 2022. In each case, an assessment of the severity of depressive disorders and an assessment of psychological test performance were conducted by the same person, i.e., a psychiatrist and a clinical psychologist, respectively.

In the DD group, HDRS scale testing was performed on the day the subjects were eligible to participate in the experiment (at most after week 2 of treatment).

In both study groups, an assessment of functioning with the ANPS and YSQ-S3-PL scales took place on the day the participants were eligible to participate in the study.

Participation in the study was voluntary, and the subjects were recruited after they had given written informed consent to participate. The study was approved by the Bioethics Committee No. RNN/136/17/KE and RNN/37/22/KE.

### 2.3. Methods of Statistical Analysis

Selected descriptive methods and methods of statistical inference were applied to analyze the data. The first step was to use descriptive statistics for all quantitative parameters of the interpreted variables. The arithmetic mean (*M*) and standard deviation (*SD*) were calculated, and the symmetry of the distribution was verified. The normality of the distribution was checked using the Shapiro–Wilk test and the Lilliefors test. They allowed the hypothesis of normality of distribution to be rejected (*p <* 0.001). The Mann–Whitney U test was used to evaluate differences between independent variables. Spearman’s rank correlation coefficient and stepwise multiple regression coefficients were used to measure the relationship between the analyzed variables. In the conducted analyses, the adopted level of significance was *p* < 0.05 [34,35]. All statistical calculations were performed using STATISTICA PL software (version 13.3).

## 3. Results

### 3.1. Neural Emotional Systems and Early Maladaptive Schemas—Differences across Study Groups

The results recorded in the ANPS and YSQ-S3-PL questionnaires in the studied group are presented in Table 3.

For the ANPS scale, the authors confirmed significant statistical differences for three neural emotional systems, namely SEEKING, PLAY, and ANGER. In each of the three cases, depressed patients scored lower than healthy subjects.

The subjects with depressive symptoms scored significantly higher in the YSQ-S3-PL questionnaire on two domains of early maladaptive schemas, i.e., “Impaired autonomy and performance” and “Other-directedness”.

### 3.2. Neural Emotional Systems and Early Maladaptive Schemas—Interrelationships

In order to look at the associations of neural emotional systems with early maladaptive schemas, a correlational analysis was conducted using Spearman’s rho correlation coefficient. Given the breadth of the data collected and because previous analyses showed significant differences between DD and CG subjects in only two domains of the YSQ-S3-PL questionnaire, the authors focused on analyses for the five domains of early maladaptive schemas (Table 4) and resigned from additional analyses for individual 18 schemas. An analysis of the entire study group (*n =* 90) was performed for the same reasons.

As indicated in Table 4, the YSQ-S3-PL total score and individual domains of early maladaptive schemas are strongly correlated with individual dimensions of the ANPS scale. Only the CARE system was found to be poorly associated with most domains of early maladaptive schemas.

The following systems are relevant for the “Disconnection and rejection” area: SEEKING, PLAY (negative correlation), ANGER, FEAR, and SADNESS (positive correlation). These links mean that individuals with impaired feelings of safety, stability, caring, and empathy in their relationships with others have an impaired tendency to seek out stimuli that provide a sense of security, a tendency to experience anger, fear, and sadness, and difficulties relating to others. Similar relationships apply to the area of “Impaired autonomy and performance”, linked to a lack of confidence in one’s own abilities and difficulties with emotional separation from significant others.

In the case of the “Impaired limits” domain, statistical significance was obtained for two emotional systems, namely CARE (negative relationship) and ANGER (positive relationship). This means that people in this group, who, according to J. Young, are characterized by difficulty in respecting the rights of others, have a strong tendency to experience feelings of anger and difficulty in showing closeness to others or accepting care from them.

The “Other-directedness” area, on the other hand, correlates with SEEKING (negative correlation), FEAR, and SADNESS (positive correlations). This means that excessive focus on other people’s emotions and desires is associated with a tendency to experience feelings of sadness and fear (these emotions are linked to fear of rejection) and a low attitude towards seeking positive sensations.

Interestingly, the last of the YSQ-S3-PL domains, namely “Overvigilance and inhibition”, centered around difficulty recognizing one’s own emotional states and over-suppressing them, is linked significantly positively to two neural emotional systems, i.e., FEAR and ANGER. This may mean that the overuse of denial and displacement mechanisms in social relationships—at the neural level—does not cancel out the emotions experienced.

### 3.3. Regression Analysis

The next step in the statistical analyses was to assess the significance of the six dimensions of the ANPS for the severity of the early maladaptive schema domains. The progressive stepwise multiple regression method was used for statistical calculations (Table 5).

The results of the regression analysis indicate that impairment of the neural emotional SEEKING system explains the variability of the “Disconnection and rejection”, “Impaired autonomy and performance”, and “Other-directedness” domains to the largest extent. For score variability in the domain area of “Impaired limits”, the ANGER system was found to be most significant, and it was the FEAR system for “Overvigilance and inhibition”.

Thus, the most important neural emotional system for patients with depressive symptoms may be the SEEKING system.

## 4. Discussion

Numerous studies confirm the association of cognitive maladaptive schemas distinguished by Jeffrey Young with the occurrence of not only symptoms of the recurrent depressive disorder but also bipolar disorder, suicidal phobia, obsessive-compulsive disorder, social phobia, addictions, and of course, personality disorders and self-harm tendencies (among others Marteinsdottir et al. [36]; Pinto-Gouveia et al. [37]; Unoka et al. [38]; Hawke and Provencher [39]; Kim et al. [40]; Kwak and Lee [41]; Khosravani et al. [42]; Munuera et al. [43]; Nicol et al. [44].

In contrast, neuroimaging studies indicate the involvement of analogous dysfunctions between the amygdala (the so-called emotional brain) and frontal lobes (the so-called rational brain) in the etiology of the same groups of disorders as mentioned above, i.e., generalized anxiety disorder (GAD) [45], borderline personality disorder [46], depressive disorders [47], bipolar affective disorder [48], substance abuse [49], or behavioral addictions [50]. However, not enough research focusing on the neurobiological basis of early maladaptive schemas has been conducted so far [51].

As with personality traits, schemas are an indispensable part of a person’s mental structure. They have the nature of unconditional beliefs, not questioned by a given person; they constitute an important part of people’s identity and their knowledge about themselves, other people, as well as about the surrounding world. Their strength, reinforcement, and frequency of activation determine the impact they have on the daily functioning of the given person [52]. A key component of early maladaptive schemas includes emotions, and differences in the expression and regulation of emotions account for a large range of individual differences in personality [53]. Following J. Panksepp, the authors treat neural emotional systems as emotional endophenotypes (emotional markers of underlying neuropsychological activity, which mediate between epigenetics and human behavior) and consider them as a component of early maladaptive schemas. These primary affective networks condition the development of higher-order mental processes and are central to the formation of an individual’s behavior and relationships when interacting with others [54,55,56].

Moreover, when faced with new challenges, different types of schemas may be activated to effectively deal with the difficult situation and the emotions that result [57]. Schemas—similarly to responses activated by neural emotional systems—are designed to help us survive in adverse environmental conditions [11].

In the authors’ opinion, investigating this issue in more detail may allow for a better understanding of the etiology and course of most mental disorders (including depression) and to link the achievements of psychology and neuroscience in the design of new therapeutic techniques [58].

### 4.1. Neurobiology of Depressive Disorders

According to Li et al. [59], dysfunctions of functional connectivity (FC) between the amygdala, the insular cortex, the anterior cingulate cortex (ACC), and the prefrontal cortex (PFC) are observed in depressive disorders. These changes primarily affect the following connections between ACC and left precuneus, ACC and left amygdala, ACC and left dorsolateral PFC, left subgenual ACC and left cerebellar, left PFC and anterior subcallosal area, and left precuneus and left pulvinar of the thalamus. Wang et al. [60], on the other hand, observed that the right anterior cingulated cortex (ACC) in depressed patients was involved not only in the regulation of emotional functions (inversely proportional relationship) but also in the normal course of executive functions (directly proportional relationship). These dysfunctions resolve to some extent as a consequence of antidepressant pharmacotherapy [59], psychotherapeutic interventions, or a series of electric shocks [61], but significant changes in the aforementioned areas are observed during subsequent episodes of the illness [62,63]. Additionally, the changes described are more severe in depressed patients who make suicide attempts compared to treated patients with the same diagnosis and no history of suicide attempts [64]. Increasingly for depressive disorders, it is emphasized that emotional and cognitive dysregulation is attributed to structural and functional abnormalities in the affective network (AN) and cognitive control network (CCN) [62].

### 4.2. Early Maladaptive Schemas and Depressive Disorders

The authors devoted their previous work to a detailed analysis of the relationship between early maladaptive schemas and depressive disorders [65]. At this point, it would be appropriate to mention the most relevant content related to this issue.

According to numerous authors, scores obtained in the early maladaptive schema questionnaire are a reliable and relatively stable marker of depressive disorders [66,67,68,69], and approximately 60% of patients with depressive symptoms achieve a symptomatic improvement following the use of psychotherapy in the form of schema therapy [70,71,72]. Kindyis et al. [73] also confirmed the effectiveness of schema therapy in alleviating depressive symptoms in older adults.

According to Cormier et al. [66], the severity of early maladaptive schemas increases with the severity of depressive symptoms, and three schemas are characteristic of its occurrence (regardless of symptoms intensity), namely “Defectiveness/shame”, “Dependence/incompetence”, and “Vulnerability to harm or illness”. In contrast, the least characteristic schema for depressed patients is “Entitlement/Grandiosity”. Interestingly, the aforementioned schemas also achieve the highest intensity in the subjects during the remission period of the disease [67], which makes it possible to consider them as indicators of vulnerability to the onset of depressive symptoms. 

The classic symptoms of depression, regardless of gender, are associated with the following domains: “Disconnection and rejection”, “Impaired autonomy and performance”, and “Other-directedness” [67,74,75]. The authors observed results consistent with those cited (areas of “Impaired autonomy and performance” and “Other-directedness” differentiated the studied groups to the largest extent).

For experiences associated with early childhood trauma, the “Disconnection and rejection” area is a particularly important moderator of depressive symptoms [76]. In a longitudinal, 9-year study involving patients with a diagnosis of depressive disorder, Wang et al. [69] found a nearly 60% correlation between the areas of “Disconnection and rejection” and “Impaired limits” and the severity of depressive symptoms. In contrast, the following schemas are associated with the risk of suicide attempts in the course of the disease: “Emotional deprivation”, “Defectiveness/shame”, “Abandonment/instability”, and “Social isolation/alienation” [77].

### 4.3. Neural Emotional Systems and Early Maladaptive Schemas in Depressive Disorders

In recent years, J. Panksepp’s theory has been linked [29] to, among others, P. Costa and T. McCrae’s BIG FIVE theory [78], Cloninger’s Temperament and Character Theory [79,80], and even A. Maslow’s theory of needs [81]. In their meta-analysis, Marengo et al. [82] found that high SEEKING relates to high Openness to Experience, high PLAY to high Extraversion, high CARE/low ANGER to high Agreeableness, and high FEAR/SADNESS/ANGER to high Neuroticism. Similar results were obtained in German [21], Spanish [20], Italian [83], and Serbian (Montag et al., 2019) studies. However, as noted earlier, there is a lack of research linking the theory of neural emotional systems to J. Young’s EMS.

According to Panksepp [84], two of the emotional systems—SADNESS and SEEKING—are particularly relevant to the development of depressive disorders. They are stimulated in case of separation from a loved one (which continues as mourning if the separation is sustained) [84]. This phase may also be characterized by significant arousal in the SEEKING system region (which is supposed to facilitate a future encounter with the beloved object). If separation is maintained, the role of the SEEKING system decreases, and the activity of the SADNESS system increases (this condition resembles depressive disorders in clinical terms) [16]. Moreover, anhedonia typical for the course of depression is associated by J. Panksepp with high SADNESS and diminished SEEKING activity. In contrast, high SEEKING activity with high SADNESS was linked by authors to the risk of engaging in self-aggressive behavior, including suicide attempts [16].

An attempt to assess the severity of individual neural emotional systems in depression was made by Montag et al. (55). A group of 669 individuals, including 55 patients diagnosed with depression, took part in the study. Those authors found a statistically significant association between lower SEEKING system scores and higher depressive tendencies among healthy individuals and those treated for depressive symptoms. Starting from this assumption, Panksepp highlights the role of overactive SEEKING in psychosis and its deficit in depression and addiction [16]. Furthermore, low SEEKING, high FEAR, and high SADNESS scores were associated with higher scores in the Beck Depression Inventory questionnaire (BDI-II) [12].

Fuchshuber et al. [85] also evaluated the relationship between the severity of depressive symptoms and subjects’ scores on the ANPS test. The cited authors indicated associations of depression with SADNESS (β = 0.53), FEAR (β = 0.10), SEEKING (β = −0.10), and PLAY (β = −0.15). Comparable results were obtained in the pre-COVID-19 study conducted by Sanwald et al. [86]. There were 44 patients treated for depression and 49 healthy controls in that project. Inpatients suffering from major depressive disorder (MDD) scored significantly lower on the primary emotion SEEKING and PLAY than controls did. Inpatients, as compared to healthy controls, had significantly higher scores with respect to FEAR and SADNESS [86]. The study was repeated during the first year of the SARS-CoV-2 pandemic (116 depressed patients and 91 healthy subjects). As a result, the primary PLAY emotion was significantly negatively associated with fear of COVID-19. Interestingly, while there was a (non-significant) positive association between SADNESS and fear of COVID-19 in the healthy controls, SADNESS was negatively associated with fear of COVID-19 in the former inpatients [86]. Sanwald et al. [87] also confirmed the increased activity of the SADNESS system with decreased activity of the SEEKING system among young women suffering from depressive disorders compared to male subjects. Montag et al. [88] also observed strong associations between higher FEAR and SADNESS scores and depressive tendencies in healthy subjects and depressed individuals.

In the presented study, the SEEKING system was also found to be most strongly associated with key early maladaptive schemas for depressive symptoms. This primary and oldest motivational system triggers actions related to the exploration of the world, interest in reality, and the search for and anticipation of positive experiences. The arousal of this system leads to intense learning processes and the production of adaptive behaviors and allows for the acquisition of knowledge [80]. Weakening of its activity would therefore imply a risk of depressive symptoms. Are we dealing with the occurrence of neural changes at the most basic level of our brain functioning in patients with symptoms of this disease? [89,90]. Answering this question certainly requires further research, but the results presented here bring us closer to an answer.

## 5. Summary

Panksepp’s model brilliantly integrates knowledge from such disparate scientific fields as neuroscience, behavioral psychology, cognitive psychology, psychoanalysis, evolutionary psychology, and attachment theory. It appears to be inspiring to both clinicians and researchers [13]. What is more, in a cross-cultural project involving 520 Canadian subjects and 830 French subjects, Orri et al. [91] demonstrated the temporal stability of personality profiles as assessed by the ANPS scale. Thus, a conclusion can be made that there is temporal stability of the human personality dimensions associated with these systems.

Despite several limitations (indicated below), this paper represents the first attempt systematically investigate the early maladaptive schemas in disease conditions. It may constitute a vital contribution to the innovation and practice of depression treatment.

## 6. Limitations

The study presented here does not provide direct evidence (e.g., in the form of functional imaging findings) to explain the mechanisms underlying the association between primary emotional systems and early maladaptive schemas. However, it allows for the formulation of hypotheses to explain the relationships obtained, which can inspire further extended research;The size of the group of depressed individuals does not allow for additional analyses that would indicate the functioning of neural emotional systems for different severity of depressive symptoms or different clinical courses of depression;Nevertheless, the work highlights the usefulness of the Affective Neuroscience Personality Scales for psychiatric diagnostics and extended studies on treatment responses in depression.

## 7. Conclusions

Two domains of early maladaptive schemas are significant for the onset of depressive symptoms, namely “Impaired autonomy and performance” and “Other-directedness”, linked to difficulties in engaging in behaviors to meet one’s own needs.

Impairment of the neural and emotional SEEKING system most significantly explains the variability in depression-typical areas of early maladaptive schemas.

## Figures and Tables

**Table 1 ijerph-19-08062-t001:** Characteristics and biological basis of Panksepp’s neuroaffective emotional systems [8,9,11,15,21,22].

System Name		Characteristics	Brain Location
**POSITIVE AFFECT**	**SEEKING** *(interest) **	The main and oldest motivational system. It stimulates activities related to the exploration of the world, interest in reality, and seeking and anticipating positive experiences. Arousal of this system leads to intense learning processes, production of adaptive behavior (basal nuclei), and knowledge (neocortex). The SEEKING neural system includes the reward system-in terms of enthusiasm and euphoria of engaging, but not hedonistic satisfaction.	It is associated with the activity of, among others, the nucleus accumbens, ventral tegmental area, lateral hypothalamic area, periaqueductal gray (PAG), as well as mesolimbic and mesocortical pathways.
	**PLAY** *(playfulness/joy)*	It controls responses related to social adaptation, formation of social patterns, and prosocial attitudes. The PLAY system is otherwise known as the physical and social engagement system. Play, as animal and human research shows, shapes social patterns that have no prior representation in the brain. PLAY reduces negative affect (e.g., anger), reinforces prosocial attitudes, influences brain neuroplasticity, and modifies the functions of other emotional systems.	The brain areas involved in this system are the dorsomedial part of the midbrain, parafascicular thalamic nucleus, and PAG.
	**CARE** *(caring/nurturance)*	It controls responses associated with maternal and nurturing behaviors and feelings and with the development of interpersonal relationships. It plays an important role in early childhood development and is linked with the activation of the opioids, oxytocin, and prolactin systems in the brain.	The areas significant for this system include, among others, the anterior cingulate cortex and bed nucleus of the stria terminalis, preoptic area, ventral tegmental area, and PAG.
**NEGATIVE AFFECT**	**ANGER** *(rage)*	It is responsible for reactions associated with experiencing feelings of anger and rage and with a tendency to exhibit aggressive behavior. The ANGER system activates when the SEEKING system is disabled.	It is located in the middle parts of the amygdala, in the bed nucleus of the stria terminalis (sometimes referred to as the extended amygdala), in the central parts of the hypothalamus, and PAG. The areas of the brain activated in RAGE include, among others, the amygdala, stria terminalis, medial hypothalamus, and PAG. This system also projects to the frontal and insular cortex.
	**FEAR** *(anxiety)*	Its activation is associated with experiencing feelings of anxiety, a tendency to worry, difficulty making decisions, frequent ruminations, and a sense of internal tension. Stimulation of this system elicits an escape or refrain response. This system is also linked to a reduction in pain sensation.	It is associated with the activity of the central and posterior amygdala, the medial part of the hypothalamus, and the dorsal part of PAG.
	**SADNESS** *(panic/separation distress/grief)*	It forms the basis of the attachment response and is activated in situations of separation from meaningful objects. It involves experiencing a sense of loneliness.	Higher SADNESS scores may be associated with lower functional connectivity between the left basolateral amygdala and the right postcentral gyrus and between the right basolateral amygdala and a so-called “subgyral” cluster in the parietal lobe (as well as the right superior parietal lobe)—see Deris et al. [11].

*—other names for particular neural emotional systems that can be found in the literature.

**Table 2 ijerph-19-08062-t002:** The social and demographic characteristics of the study group.

Variables		DD *n* = 45		CG *n* = 45		All Subjects *n* = 90	
		*M*	*SD*	*M*	*SD*	*M*	*SD*
Age (years)		41.26	11.68	21.52	2.02	31.39	12.97
Severity of depressive disorder symptoms as measured by HDRS		25.07	7.91	-		-	
Gender	Females	*N*	*%*	*N*	*%*	*N*	*%*
		31	68.89	31	68.89	62	68.89
	Males	14	31.11	14	31.11	28	31.11

DD—depressive disorder group; CG—comparison group; N—size; M—mean; SD—standard deviation; %—percentage; HDRS—Hamilton Depression Rating Scale.

**Table 3 ijerph-19-08062-t003:** Neural emotional systems and early maladaptive schemas across the studied groups.

Variables	DD *n* = 45				GC *n* = 45				All Subjects *n* = 90				DD v GC	
	*M*	*SD*	*Min*	*Max.*	*M*	*SD*	*Min*	*Max.*	*M*	*SD*	*Min*	*Max*	*Mann*–*Whitney U test*	*p*
**ANPS**														
SEEKING	21.04	7.16	5	39	27.93	6.01	11	42	24.49	7.43	5	42	469.51	0.001 *
PLAY	19.89	6.24	7	30	26.98	8.08	9	40	23.43	8.01	7	40	478.51	0.001 *
CARE	24.18	7.22	7	42	26.61	7.01	6	38	25.39	7.18	6	42	789.01	0.071
ANGER	17.27	6.29	1	27	22.42	7.62	6	42	19.84	7.42	1	42	652.51	0.003 **
FEAR	23.47	6.06	12	39	25.78	7.94	7	42	24.62	7.12	7	42	808.01	0.09
SADNESS	22.51	6.31	11	39	22.49	7.45	2	37	22.51	6.86	2	39	980.51	0.791
**YSQ-S3-PL**														
Sum	222.71	118.24	31	436	241.36	56.91	122	343	232.58	90.98	31	436	829.51	0.537
*DISCONNECTION AND REJECTION*														
Sum	76.88	36.79	30	144	64.31	21.22	26	106	69.53	29.18	26	144	605.51	0.238
Emotional deprivation	14.66	8.27	5	29	10.42	5.49	5	25	12.18	7.05	5	29	503.51	0.021 **
Abandonment/instability	17.47	6.79	7	30	16.07	6.56	5	29	16.65	6.65	5	29	644.51	0.438
Mistrust/abuse	15.37	7.93	5	29	14.04	5.77	5	23	14.61	6.74	5	30	672.01	0.623
Defectiveness/shame	14.47	8.16	5	30	10.41	6.13	5	26	12.09	7.28	5	30	515.01	0.034 **
Social isolation/alienation	14.91	7.79	5	28	13.38	6.06	5	29	14.01	6.82	5	29	651.51	0.482
*IMPAIRED AUTONOMY AND PERFORMANCE*														
Sum	58.59	26.33	23	107	43.89	15.56	20	74	50.01	21.82	20	107	508.51	0.029 **
Dependence/incompetence	14.72	6.97	5	26	9.61	3.29	5	17	11.73	5.71	5	26	434.01	0.003 **
Vulnerability to harm														
or illness	14.44	6.86	5	28	13.24	6.08	5	26	13.74	6.41	5	28	662.01	0.552
Enmeshment/undeveloped self	13.94	7.78	5	28	9.76	5.09	5	25	11.49	6.63	5	28	523.01	0.042 **
Failure to achieve	15.51	7.76	5	30	11.29	4.73	5	23	13.04	6.47	5	30	511.51	0.031 **
*IMPAIRED LIMITS*														
Sum	29.22	11.85	12	53	28.47	6.78	24	72	28.78	9.17	12	72	690.51	0.764
Entitlement/grandiosity	13.63	6.71	6	26	15.16	3.67	9	26	14.52	5.17	6	26	587.01	0.171
Insufficient self-control/self-discipline	15.59	5.92	5	29	13.31	5.32	5	27	14.26	5.78	5	29	542.01	0.067
*OTHER-DIRECTEDNESS*														
Sum	52.94	13.36	36	82	45.64	10.25	24	72	48.68	12.11	24	82	508.01	0.028 **
Subjugation	16.94	6.96	7	29	11.22	3.85	5	21	13.61	6.03	5	29	396.51	0.001 *
Self-sacrifice	18.69	5.13	9	28	16.53	5.19	8	28	17.43	5.24	8	28	546.01	0.072
Approval-seeking/recognition-seeking	17.31	4.71	7	28	17.89	5.08	9	28	17.65	4.91	7	28	662.51	0.555
*OVERVIGILANCE AND INHIBITION*														
Sum	65.22	22.31	28	108	59.94	13.91	30	86	60.55	18.87	28	108	610.01	0.357
Negativity/pessimism	16.56	7.01	5	28	15.11	5.56	5	27	15.71	6.21	5	28	630.01	0.354
Emotional inhibition	16.11	6.29	6	27	13.09	5.19	5	26	14.31	5.82	5	27	521.51	0.041 **
Unrelenting standards	17.16	5.01	9	17	17.71	4.75	8	27	17.48	4.83	8	27	686.01	0.729
Punitiveness	15.51	6.62	5	27	13.13	4.54	5	23	14.12	5.59	5	27	573.01	0.131

ANPS—Affective Neuroscience Personality Scale; YSQ-S3-PL—Young Schema Questionnaire; DD—depressive disorders; CG—comparison group; N—size; M—mean; SD—standard deviation; *—*p* ≤ 0.001; **—*p* ≤ 0.05.

**Table 4 ijerph-19-08062-t004:** Values of Spearman’s rho correlation coefficient for the ANPS scale scores and the five domains of the YSQ-S3-PL questionnaire for the study group (*n* = 90).

	All Subjects *n* = 90											
	ANPS											
	SEEKING		PLAY		CARE		ANGER		FEAR		SADNESS	
	*R*	*p*	*R*	*p*	*R*	*p*	*R*	*p*	*R*	*p*	*R*	*p*
**YSQ-S3-PL Sum**	−0.224	0.039 **	−0.087	0.428	−0.173	0.133			0.428	0.001 *	0.348	0.001 *
** *Disconnection and rejection* **	−0.546	0.001 *	−0.448	0.001 *	0.003	0.979	0.314	0.005 **	0.511	0.001 *	0.464	0.001 *
** *Impaired autonomy and performance* **	−0.548	0.001 *	−0.378	0.001 *	−0.0176	0.126	0.301	0.008 **	0.516	0.001 *	0.528	0.001 *
** *Impaired limits* **	−0.217	0.059	0.036	0.756	−0.325	0.004 **	0.594	0.001 *	0.136	0.237	0.061	0.603
** *Other-directedness* **	−0.448	0.001 *	−0.207	0.071	0.076	0.514	0.192	0.094	0.457	0.001 *	0.528	0.001 *
** *Overvigilance and inhibition* **	−0.105	0.364	−0.101	0.384	−0.068	0.557	0.233	0.043 **	0.264	0.021 **	0.186	0.107

ANPS—Affective Neuroscience Personality Scale; YSQ-S3-PL—Young Schema Questionnaire; *—*p* ≤ 0.001; **—*p* ≤ 0.05.

**Table 5 ijerph-19-08062-t005:** Progressive stepwise regression coefficient results for the five domains of the YSQ-S3-PL questionnaire (dependent variables) and for the ANPS dimensions (independent variables) for the entire study group (*n* = 90).

Variable	YSQ-S3-PL		
	*Disconnection and Rejection*		
	*R^2^*	*b*	*p*
**Absolute term**		15.794	
**SEEKING**		−0.381	
**SADNESS**		0.241	
**ANGER**		0.252	
**PLAY**	0.437	−0.211	0.001 *
	* **Impaired autonomy and performance** *		
	** *R^2^* **	** *b* **	** *p* **
**Absolute term**		12.287	
**SEEKING**		−0.481	
**ANGER**	0.394	0.257	0.001 *
**SADNESS**		0.205	
	* **Impaired limits** *		
	** *R^2^* **	** *b* **	** *p* **
**Absolute term**		5.778	
**PLAY**		0.164	
**SEEKING**		−0.371	
**CARE**		−0.081	
**FEAR**		−0.111	
**ANGER**	0.372	0.534	0.001 *
	* **Other-directedness** *		
	** *R^2^* **	** *b* **	** *p* **
**Absolute term**	0.339	7.676	0.001 *
**SEEKING**		−0.571	
**FEAR**		0.061	
**CARE**		0.305	
**ANGER**		0.262	
	* **Overvigilance and inhibition** *		
	** *R^2^* **	** *b* **	** *p* **
**Absolute term**		13.03	
**PLAY**		−0.021	
**SEEKING**		0.041	
**CARE**		−0.14	
**FEAR**	0.103	0.327	0.001 *
**ANGER**		0.021	

ANPS—Affective Neuroscience Personality Scale; YSQ-S3-PL—Young Schema Questionnaire; *—*p* ≤ 0.001; gray color indicates strongest relationships.

## Data Availability

Not applicable.

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
