# Peer review of "Affective Neuroscience Personality Scales and Early Maladaptive Schemas in Depressive Disorders"

_ijerph, 2022, doi:10.3390/ijerph19138062_

Round 1

Reviewer 1 Report

I thank the authors for presenting their study. However, adjustments are necessary for consideration:

  • The citation system must be adjusted according to the standards of the IJERP.
  • Please clarify the study design, to understand how the process was to recruit the 90 participants. The inclusion criteria for the DD are clear, but not for the CG.
  • Add the characteristics of the sample, for example, mean age and SD.
  • The discussion must be refromulated to contrast the results obtaine in their study with what exist in the literature.
  • It is necessary to order the presentation of the manuscript. For example, the limitations should be before the conclusion (follow the format of a traditional original study).

Author Response

Reviewer 1.

Thank you for preparing the review. The introduced changes are marked in the text.

Regarding the following comments:

Ad. 1.

The citation system must be adjusted according to the standards of the IJERP.

Citations have been corrected.

Ad 2.

Please clarify the study design, to understand how the process was to recruit the 90 participants. The inclusion criteria for the DD are clear, but not for the CG.

The description of the CG group was supplemented.

Ad 3.

Add the characteristics of the sample, for example, mean age and SD.

The social and demographic characteristics of the study group are presented in Table 2.

Ad 4.

The discussion must be refromulated to contrast the results obtaine in their study with what exist in the literature.

However, we cannot indicate any differences between our study and others, because it is the first study to show the analyzed dependencies (ANPS and YSQ scale in the group of people with depression).

Ad 5.

It is necessary to order the presentation of the manuscript. For example, the limitations should be before the conclusion (follow the format of a traditional original study).

We applied the reviewer's comment.

Reviewer 2 Report

The article is well-structured, organized and the questions are precisely constructed. Introduction, methodology and other sections are easy to understood however the clinical significance and implementation into the practise or further investigation is dificult to comprehend. Within this, it is recommended to describe the topic within the bigger picture. Otherwise the article brings scientific problematic with proper methodology used.

Table 1 contains sentences which appears cut in the middle. Sentence from the line 103-106 is difficult to understand and requires reediting. Sentence in the line 138 does not have an ending.

Author Response

Reviewer 2.

Thank you for preparing the review. The introduced changes are marked in the text.

Regarding the following comments:

Ad 1.
“The article is well-structured, organized and the questions are precisely constructed. Introduction, methodology and other sections are easy to understood however the clinical significance and implementation into the practise or further investigation is dificult to comprehend. Within this, it is recommended to describe the topic within the bigger picture. Otherwise the article brings scientific problematic with proper methodology used.

“Until the moment of compiling this article, the authors did not confirm the pres-ence of other studies linking Jeffrey Young's theory of early maladaptive schemas with J. Panksepp's Affective Neuroscience Personality Theory. The study presented herein is therefore the first attempt to deal with this matter. It is hoped that the results obtained will lead to a better understanding of the biological basis of early maladaptive schemas and, in the future, will contribute to an increase in the effectiveness of applied thera-peutic interventions”.

The comments mentioned by the Reviewer are included in the “Aim of the study” section.

Ad 2.
Table 1 contains sentences which appears cut in the middle. Sentence from the line 103-106 is difficult to understand and requires reediting. Sentence in the line 138 does not have an ending”.

Corrections were made in the indicated fragments of the article.

Reviewer 3 Report

This article is of high quality, and presents both accurate and important topic to be raised and careful planning of the study.

There are some minor remarks I wish to note:

1. Maybe refer to (I think it will be the MOST infulential theory in both in neuroscience and psychology of emotions yet to be) constructivist theory of emotions. As a cultural psychologists I am more than happy to hear that its getting more and more popular among neuroscientists these days and has a strong basis in neroscientifical data. See e.g. the basic reading

https://lisafeldmanbarrett.com/books/how-emotions-are-made/

2. There is overrepresentation of women within the study. I am aware that women are diagnosed with depression more often as males are mis/diagnosed more often with addiction (and the other way round  certainly that women are often mis/diagnosed less with addiction)

3. I do not venture into statistics as I am a qualitatively-oriented psychologists. So I suppose the math is fine. From my point of view with such a small group interviews would prove a more accurate method but licentia poetica through.

Author Response

Reviewer 3.

Thank you for preparing the review. The introduced changes are marked in the text.

Regarding the following comments:

A1.
“Maybe refer to (I think it will be the MOST infulential theory in both in neuroscience and psychology of emotions yet to be) constructivist theory of emotions. As a cultural psychologists I am more than happy to hear that its getting more and more popular among neuroscientists these days and has a strong basis in neroscientifical data. See e.g. the basic reading

https://lisafeldmanbarrett.com/books/how-emotions-are-made/”.

Thank you for your kind words.

Due to the length of the paper, we decided to keep it in its current shape.

Ad2.

“There is overrepresentation of women within the study. I am aware that women are diagnosed with depression more often as males are mis/diagnosed more often with addiction (and the other way round  certainly that women are often mis/diagnosed less with addiction)”.

Ad3.
“I do not venture into statistics as I am a qualitatively-oriented psychologists. So I suppose the math is fine. From my point of view with such a small group interviews would prove a more accurate method but licentia poetica through”.

We are aware of the advantage of women in the study group, however, statistical analyzes were performed according to the applicable standards.

Round 2

Reviewer 1 Report

Thank you for the adjustments!!!